# Anytime Influence Bounds and the Explosive Behavior of Continuous-Time Diffusion Networks

Kevin Scaman[1]      Rémi Lemonnier[1,2]      Nicolas Vayatis[1]

[1]CMLA, ENS Cachan, CNRS, Université Paris- Saclay, France, [2]1000mercis, Paris, France

`{scaman, lemonnier, vayatis}@cmla.ens-cachan.fr`

## Abstract

The paper studies transition phenomena in information cascades observed along a diffusion process over some graph. We introduce the Laplace Hazard matrix and show that its spectral radius fully characterizes the dynamics of the contagion both in terms of influence and of explosion time. Using this concept, we prove tight non-asymptotic bounds for the influence of a set of nodes, and we also provide an in-depth analysis of the critical time after which the contagion becomes super-critical. Our contributions include formal definitions and tight lower bounds of critical explosion time. We illustrate the relevance of our theoretical results through several examples of information cascades used in epidemiology and viral marketing models. Finally, we provide a series of numerical experiments for various types of networks which confirm the tightness of the theoretical bounds.

## 1   Introduction

Diffusion networks capture the underlying mechanism of how events propagate throughout a complex network. In marketing, social graph dynamics have caused large transformations in business models, forcing companies to re-imagine their customers not as a mass of isolated economic agents, but as *customer networks* [1]. In epidemiology, a precise understanding of spreading phenomena is heavily needed when trying to break the chain of infection in populations during outbreaks of viral diseases. But whether the subject is a virus spreading across a computer network, an innovative product among early adopters, or a rumor propagating on a network of people, the questions of interest are the same: how many people will it infect? How fast will it spread? And, even more critically for decision makers: how can we modify its course in order to meet specific goals? Several papers tackled these issues by studying the *influence maximization* problem. Given a known diffusion process on a graph, it consists in finding the top-k subset of initial seeds with the highest expected number of infected nodes at a certain time distance $T$. This problem being NP-hard [2], various heuristics have been proposed in order to obtain scalable suboptimal approximations. While the first algorithms focused on discrete-time models and the special case $T = +\infty$ [3, 4], subsequent papers [5, 6] brought empirical evidences of the key role played by temporal behavior. Existing models of continuous-time stochastic processes include multivariate Hawkes processes [7] where recent progress in inference methods [8, 9] made available the tools for the study of activity shaping [10], which is closely related to influence maximization. However, in the most studied case in which each node of the network can only be infected once, the most widely used model remains the Continuous-Time Information Cascade (CTIC) model [5]. Under this framework, successful inference [5] as well as influence maximization algorithms have been developed [11, 12].

However, if recent works [13, 14] provided theoretical foundations for the inference problem, assessing the quality of influence maximization remains a challenging task, as few theoretical results exist for general graphs. In the infinite-time setting, studies of the SIR diffusion process in epidemiology [15] or percolation for specific graphs [16] provided a more accurate understanding of these processes. More recently, it was shown in [17] that the spectral radius of a given *Hazard matrix*

played a key role in influence of information cascades. This allowed the authors to derive closed-form tight bounds for the influence in general graphs and characterize *epidemic thresholds* under which the influence of any set of nodes is at most $O(\sqrt{n})$.

In this paper, we extend their approach in order to deal with the problem of *anytime influence bounds* for continuous-time information cascades. More specifically, we define the *Laplace Hazard matrices* and show that the influence at time $T$ of any set of nodes heavily depends on their spectral radii. Moreover, we reveal the existence and characterize the behavior of *critical times* at which super-critical processes explode. We show that before these times, super-critical processes will behave sub-critically and infect at most $o(n)$ nodes. These results can be used in various ways. First, they provide a way to evaluate influence maximization algorithms without having to test all possible set of influencers, which is intractable for large graphs. Secondly, critical times allow decision makers to know how long a contagion will remain in its early phase before becoming a large-scale event, in fields where knowing *when* to act is nearly as important as knowing *where* to act. Finally, they can be seen as the first closed-form formula for anytime influence estimation for continuous-time information cascades. Indeed, we provide empirical evidence that our bounds are tight for a large family of graphs at the beginning and the end of the infection process.

The rest of the paper is organized as follows. In Section 2, we recall the definition of Information Cascades Model and introduce useful notations. In Section 3, we derive theoretical bounds for the influence. In Section 4, we illustrate our results by applying them on specific cascade models. In Section 5, we perform experiments in order to show that our bounds are sharp for a family of graphs and sets of initial nodes. All proof details are provided in the supplementary material.

## 2 Continuous-Time Information Cascades

### 2.1 Information propagation and influence in diffusion networks

We describe here the propagation dynamics introduced in [5]. Let $\mathcal{G} = (\mathcal{V}, \mathcal{E})$ be a directed network of $n$ nodes. We equip each directed edge $(i, j) \in \mathcal{E}$ with a time-varying probability distribution $p_{ij}(t)$ over $\mathbb{R}_+ \cup \{+\infty\}$ ($p_{ij}$ is thus a sub-probability measure on $\mathbb{R}_+$) and define the cascade behavior as follows. At time $t = 0$, only a subset $A \subset \mathcal{V}$ of *influencers* is infected. Each node $i$ infected at time $\tau_i$ may transmit the infection at time $\tau_i + \tau_{ij}$ along its outgoing edge $(i, j) \in \mathcal{E}$ with probability density $p_{ij}(\tau_{ij})$, and independently of other transmission events. The process ends for a given $T > 0$.

For each node $v \in \mathcal{V}$, we will denote as $\tau_v$ the (possibly infinite) time at which it is reached by the infection. The *influence* of $A$ at time $T$, denoted as $\sigma_A(T)$, is defined as the expected number of nodes reached by the contagion at time $T$ originating from $A$, i.e.

$$\sigma_A(T) = \mathbb{E}[\sum_{v \in \mathcal{V}} \mathbb{1}_{\{\tau_v \leq T\}}], \tag{1}$$

where the expectation is taken over cascades originating from $A$ (i.e. $\tau_v = 0 \Leftrightarrow \mathbb{1}_{\{v \in A\}}$).

Following the percolation literature, we will differentiate between *sub-critical* cascades whose size is $o(n)$ and *super-critical* cascades whose size is proportional to $n$, where $n$ denotes the size of the network. This work focuses on upper bounding the influence $\sigma_A(T)$ for any given time $T$ and characterizing the critical times at which phase transitions occur between sub-critical and super-critical behaviors.

### 2.2 The Laplace Hazard Matrix

We extend here the concept of *hazard matrix* first introduced in [17] (different from the homonym notion of [13]), which plays a key role in the influence of the information cascade.

**Definition 1.** *Let $\mathcal{G} = (\mathcal{V}, \mathcal{E})$ be a directed graph, and $p_{ij}$ be integrable edge transmission probabilities such that $\int_0^{+\infty} p_{ij}(t)dt < 1$. For $s \geq 0$, let $\mathcal{LH}(s)$ be the $n \times n$ matrix, denoted as the* Laplace hazard matrix*, whose coefficients are*

$$\mathcal{LH}_{ij}(s) = \begin{cases} -\hat{p}_{ij}(s) \left( \int_0^{+\infty} p_{ij}(t)dt \right)^{-1} \ln \left( 1 - \int_0^{+\infty} p_{ij}(t)dt \right) & \text{if } (i,j) \in \mathcal{E} \\ 0 & \text{otherwise} \end{cases}. \tag{2}$$

where $\hat{p}_{ij}(s)$ denotes the Laplace transform of $p_{ij}$ defined for every $s \geq 0$ by $\hat{p}_{ij}(s) = \int_0^{+\infty} p_{ij}(t)e^{-st}dt$. Note that the long term behavior of the cascade is retrieved when $s = 0$ and coincides with the concept of hazard matrix used in [17].

We recall that for any square matrix $M$ of size $n$, its spectral radius $\rho(M)$ is the maximum of the absolute values of its eigenvalues. If $M$ is moreover real and positive, we also have $\rho(\frac{M+M^\top}{2}) = \sup_{x \in \mathbb{R}^n} \frac{x^\top M x}{x^\top x}$.

### 2.3 Existence of a critical time of a contagion

In the following, we will derive critical times before which the contagion is sub-critical, and above which the contagion is super-critical. We now formalize this notion of critical time via limits of contagions on networks.

**Theorem 1.** *Let $(\mathcal{G}_n)_{n \in \mathbb{N}}$ be a sequence of networks of size $n$, and $(p_{ij}^n)_{n \in \mathbb{N}}$ be transmission probability functions along the edges of $\mathcal{G}_n$. Let also $\sigma_n(t)$ be the maximum influence in $\mathcal{G}_n$ at time $t$ from a single influencer. Then there exists a* critical time $T^c \in \mathbb{R}_+ \cup \{+\infty\}$ *such that, for every sequence of times $(T_n)_{n \in \mathbb{N}}$:*

- *If $\limsup_{n \to +\infty} T_n < T^c$, then $\sigma_n(T_n) = o(n)$,*
- *If $\sigma_n(T_n) = o(n)$, then $\liminf_{n \to +\infty} T_n \leq T^c$.*

*Moreover, such a critical time is unique.*

In other words, the *critical time* is a time before which the regime is *sub-critical* and after which no contagion can be *sub-critical*. The next proposition shows that, after the critical time, the contagion is *super-critical*.

**Proposition 1.** *If $(T_n)_{n \in \mathbb{N}}$ is such that $\liminf_{n \to +\infty} T_n > T^c$, then $\liminf_{n \to +\infty} \frac{\sigma_n(T_n)}{n} > 0$ and the contagion is* super-critical. *Conversely, if $(T_n)_{n \in \mathbb{N}}$ is such that $\liminf_{n \to +\infty} \frac{\sigma_n(T_n)}{n} > 0$, then $\limsup_{n \to +\infty} T_n \geq T^c$.*

In order to simplify notations, we will omit in the following the dependence in $n$ of all the variables whenever stating results holding in the limit $n \to +\infty$.

## 3 Theoretical bounds for the influence of a set of nodes

We now present our upper bounds on the influence at time $T$ and derive a lower bound on the critical time of a contagion.

### 3.1 Upper bounds on the maximum influence at time $T$

The next proposition provides an upper bound on the influence at time $T$ for any set of influencers $A$ such that $|A| = n_0$. This result may be valuable for assessing the quality of influence maximization algorithms in a given network.

**Proposition 2.** *Define $\rho(s) = \rho(\frac{\mathcal{LH}(s) + \mathcal{LH}(s)^\top}{2})$. Then, for any $A$ such that $|A| = n_0 < n$, denoting by $\sigma_A(T)$ the expected number of nodes reached by the cascade starting from $A$ at time $T$:*

$$\sigma_A(T) \leq n_0 + (n - n_0) \min_{s \geq 0} \gamma(s)e^{sT}. \tag{3}$$

*where $\gamma(s)$ is the smallest solution in $[0, 1]$ of the following equation:*

$$\gamma(s) - 1 + \exp\left(-\rho(s)\gamma(s) - \frac{\rho(s)n_0}{\gamma(s)(n - n_0)}\right) = 0. \tag{4}$$

**Corollary 1.** *Under the same assumptions:*

$$\sigma_A(T) \leq n_0 + \sqrt{n_0(n-n_0)} \min_{\{s \geq 0 | \rho(s) < 1\}} \left( \sqrt{\frac{\rho(s)}{1-\rho(s)}} e^{sT} \right), \tag{5}$$

Note that the long-term upper bound in [17] is a corollary of Proposition 2 using $s = 0$. When $\rho(0) < 1$, Corollary 1 with $s = 0$ implies that the regime is sub-critical for all $T \geq 0$. When $\rho(0) \geq 1$, the long-term behavior may be super-critical and the influence may reach linear values in $n$. However, at a cost growing exponentially with $T$, it is always possible to choose a $s$ such that $\rho(s) < 1$ and retrieve a $O(\sqrt{n})$ behavior. While the exact optimal parameter $s$ is in general not explicit, two choices of $s$ derive relevant results: either simplifying $e^{sT}$ by choosing $s = 1/T$, or keeping $\gamma(s)$ sub-critical by choosing $s$ s.t. $\rho(s) < 1$. In particular, the following corollary shows that the contagion explodes at most as $e^{\rho^{-1}(1-\epsilon)T}$ for any $\epsilon \in [0, 1]$.

**Corollary 2.** *Let $\epsilon \in [0, 1]$ and $\rho(0) \geq 1$. Under the same assumptions:*

$$\sigma_A(T) \leq n_0 + \sqrt{\frac{n_0(n-n_0)}{\epsilon}} e^{\rho^{-1}(1-\epsilon)T}. \tag{6}$$

**Remark.** Since this section focuses on bounding $\sigma_A(T)$ for a given $T \geq 0$, all the aforementioned results also hold for $p_{ij}^T(t) = p_{ij}(t)\mathbb{1}_{\{t \leq T\}}$. This is equivalent to integrating everything on $[0, T]$ instead of $\mathbb{R}_+$, i.e. $\mathcal{LH}_{ij}(s) = -\ln(1 - \int_0^T p_{ij}(t)dt)(\int_0^T p_{ij}(t)dt)^{-1} \int_0^T p_{ij}(t)e^{-st}dt$. This choice of $\mathcal{LH}$ is particularly useful when some edges are transmitting the contagion with probability 1, see for instance the SI epidemic model in Section 4.3).

## 3.2 Lower bound on the critical time of a contagion

The previous section presents results about how explosive a contagion is. These findings suggest that the speed at which a contagion explodes is bounded by a certain quantity, and thus that the process needs a certain amount of time to become super-critical. This intuition is made formal in the following corollary:

**Corollary 3.** *Assume $\forall n \geq 0, \rho_n(0) \geq 1$ and $\lim_{n \to +\infty} \frac{\rho_n^{-1}(1-\frac{1}{\ln n})}{\rho_n^{-1}(1)} = 1$. If the sequence $(T_n)_{n \in \mathbb{N}}$ is such that*

$$\limsup_{n \to +\infty} \frac{2\rho_n^{-1}(1)T_n}{\ln n} < 1. \tag{7}$$

*Then,*

$$\sigma_A(T_n) = o(n). \tag{8}$$

In other words, the regime of the contagion is *sub-critical* before $\frac{\ln n}{2\rho_n^{-1}(1)}$ and

$$T^c \geq \liminf_{n \to +\infty} \frac{\ln n}{2\rho_n^{-1}(1)}. \tag{9}$$

The technical condition $\lim_{n \to +\infty} \frac{\rho_n^{-1}(1-\frac{1}{\ln n})}{\rho_n^{-1}(1)} = 1$ imposes that, for large $n$, $\lim_{\epsilon \to 0} \frac{\rho_n^{-1}(1-\epsilon)}{\rho_n^{-1}(1)}$ converges sufficiently fast to 1 so that $\rho_n^{-1}(1 - \frac{1}{\ln n})$ has the same behavior than $\rho_n^{-1}(1)$. This condition is not very restrictive, and is met for the different case studies considered in Section 4.

This result may be valuable for decision makers since it provides a safe time region in which the contagion has not reached a macroscopic scale. It thus provides insights into *how long* do decision makers have to prepare control measures. After $T^c$, the process can explode and immediate action is required.

## 4 Application to particular contagion models

In this section, we provide several examples of cascade models that show that our theoretical bounds are applicable in a wide range of scenarios and provide the first results of this type in many areas, including two widely used epidemic models.

## 4.1 Fixed transmission pattern

When the transmission probabilities are of the form $p_{ij}(t) = \alpha_{ij}p(t)$ s.t. $\int_0^{+\infty} p(t) = 1$ and $\alpha_{ij} < 1$,

$$\mathcal{LH}_{ij}(s) = -\ln(1-\alpha_{ij})\hat{p}(s), \tag{10}$$

and

$$\rho(s) = \rho_\alpha \hat{p}(s), \tag{11}$$

where $\rho_\alpha = \rho(0) = \rho(-\frac{\ln(1-\alpha_{ij})+\ln(1-\alpha_{ji})}{2})$ is the long-term hazard matrix defined in [17]. In these networks, the temporal and structural behaviors are clearly separated. While $\rho_\alpha$ summarizes the structure of the network and how connected the nodes are to one another, $\hat{p}(s)$ captures how fast the transmission probabilities are fading through time.

When $\rho_\alpha \geq 1$, the long-term behavior is super-critical and the bound on the critical times is given by inverting $\hat{p}(s)$

$$T^c \geq \liminf_{n\to+\infty} \frac{\ln n}{2\hat{p}^{-1}(1/\rho_\alpha)}, \tag{12}$$

where $\hat{p}^{-1}(1/\rho_\alpha)$ exists and is unique since $\hat{p}(s)$ is decreasing from 1 to 0. In general, it is not possible to give a more explicit version of the critical time of Corollary 3, or of the anytime influence bound of Proposition 2. However, we investigate in the rest of this section specific $p(t)$ which lead to explicit results.

## 4.2 Exponential transmission probabilities

A notable example of fixed transmission pattern is the case of exponential probabilities $p_{ij}(t) = \alpha_{ij}\lambda e^{-\lambda t}$ for $\lambda > 0$ and $\alpha_{ij} \in [0,1[$. Influence maximization algorithms under this specific choice of transmission functions have been for instance developed in [11]. In such a case, we can calculate the spectral radii explicitly:

$$\rho(s) = \frac{\lambda}{s+\lambda}\rho_\alpha, \tag{13}$$

where $\rho_\alpha = \rho(-\frac{\ln(1-\alpha_{ij})+\ln(1-\alpha_{ji})}{2})$ is again the long-term hazard matrix. When $\rho_\alpha > 1$, this leads to a critical time lower bounded by

$$T^c \geq \liminf_{n\to+\infty} \frac{\ln n}{2\lambda(\rho_\alpha - 1)}. \tag{14}$$

The influence bound of Corollary 1 can also be reformulated in the following way:

**Corollary 4.** *Assume $\rho_\alpha \geq 1$, or else $\lambda T(1-\rho_\alpha) < \frac{1}{2}$. Then the minimum in Eq. 5 is met for $s = \frac{1}{2T} + \lambda(\rho_\alpha - 1)$ and Corollary 1 rewrites:*

$$\sigma_A(T) \leq n_0 + \sqrt{n_0(n-n_0)}\sqrt{2eT\lambda\rho_\alpha}e^{\lambda T(\rho_\alpha - 1)}. \tag{15}$$

*If $\rho_\alpha < 1$ and $\lambda T(1-\rho_\alpha) \geq \frac{1}{2}$, the minimum in Eq. 5 is met for $s = 0$ and Corollary 1 rewrites:*

$$\sigma_A(T) \leq n_0 + \sqrt{n_0(n-n_0)}\sqrt{\frac{\rho_\alpha}{1-\rho_\alpha}}. \tag{16}$$

Note that, in particular, the condition of Corollary 4 is always met in the super-critical case where $\rho_\alpha > 1$. Moreover, we retrieve the $O(\sqrt{n})$ behavior when $T < \frac{1}{\lambda(\rho_\alpha-1)}$. Concerning the behavior in $T$, the bound matches exactly the infinite-time bound when $T$ is very large in the sub-critical case. However, for sufficiently small $T$, we obtain a greatly improved result with a very instructive growth in $O(\sqrt{T})$.

## 4.3 SI and SIR epidemic models

Both epidemic models SI and SIR are particular cases of exponential transmission probabilities. SIR model ([18]) is a widely used epidemic model that uses three states to describe the spread of an infection. Each node of the network can be either : susceptible (S), infected (I), or removed (R). At

$t = 0$, a subset $A$ of $n_0$ nodes is infected. Then, each node $i$ infected at time $\tau_i$ is removed at an exponentially-distributed time $\theta_i$ of parameter $\delta$. Transmission along its outgoing edge $(i, j) \in \mathcal{E}$ occurs at time $\tau_i + \tau_{ij}$ with conditional probability density $\beta \exp(-\beta \tau_{ij})$, given that node $i$ has not been removed at that time. When the removing events are not observed, SIR is equivalent to $CTIC$, except that transmission along outgoing edges of one node are positively correlated. However, our results still hold in case of such a correlation, as shown in the following result.

**Proposition 3.** *Assume the propagation follow a SIR model of transmission parameter $\beta$ and removal parameter $\delta$. Define $p_{ij}(t) = \beta \exp(-(\delta + \beta)t)$ for $(i, j) \in \mathcal{E}$. Let $\mathcal{A} = \left(\mathbb{1}_{\{(i,j) \in \mathcal{E}\}}\right)_{ij}$ be the adjacency matrix of the underlying undirected network. Then, results of Proposition 2 and subsequent corollaries still hold with $\rho(s)$ given by:*

$$\rho(s) = \rho\left(\frac{\mathcal{LH}(s) + \mathcal{LH}(s)^\top}{2}\right) = \ln\left(1 + \frac{\beta}{\delta}\right)\frac{\delta + \beta}{s + \delta + \beta}\rho(\mathcal{A}) \tag{17}$$

From this proposition, the same analysis than in the independent transmission events case can be derived, and the critical time for the SIR model is

$$T^c \geq \liminf_{n \to +\infty} \frac{\ln n}{2(\delta + \beta)(\ln(1 + \frac{\beta}{\delta})\rho(\mathcal{A}) - 1)}. \tag{18}$$

**Proposition 4.** *Consider the SIR model with transmission rate $\beta$, recovery rate $\delta$ and adjacency matrix $\mathcal{A}_n$. Assume $\liminf_{n \to +\infty} \ln(1 + \frac{\beta}{\delta})\rho(\mathcal{A}_n) > 1$, and the sequence $(T_n)_{n \in \mathbb{N}}$ is such that*

$$\limsup_{n \to +\infty} \frac{2(\delta + \beta)(\ln(1 + \frac{\beta}{\delta})\rho(\mathcal{A}_n) - 1)T_n}{\ln n} < 1. \tag{19}$$

*Then,*

$$\sigma_A(T_n) = o(n). \tag{20}$$

This is a direct corollary of Corollary 3 with $\rho^{-1}(1) = (\delta + \beta)(\ln(1 + \frac{\beta}{\delta})\rho(\mathcal{A}_n) - 1)$.

The SI model is a simpler model in which individuals of the network remain infected and contagious through time (i.e. $\delta = 0$). Thus, the network is totally infected at the end of the contagion and $\lim_{n \to +\infty} \sigma_A(T) = n$. For this reason, the previous critical time for the more general SIR model is of no use here, and a more precise analysis is required. Following the remark of Section 3.1, we can integrate $p_{ij}$ on $[0, T]$ instead of $\mathbb{R}_+$, which leads to the following result:

**Proposition 5.** *Consider the SI model with transmission rate $\beta$ and adjacency matrix $\mathcal{A}_n$. Assume $\liminf_{n \to +\infty} \rho(\mathcal{A}_n) > 0$ and the sequence $(T_n)_{n \in \mathbb{N}}$ is such that*

$$\limsup_{n \to +\infty} \frac{\beta T_n}{\sqrt{\frac{\ln n}{2\rho(\mathcal{A}_n)}}(1 - e^{-\sqrt{\frac{\ln n}{2\rho(\mathcal{A}_n)}}})} < 1. \tag{21}$$

*Then,*

$$\sigma_A(T_n) = o(n). \tag{22}$$

In other words, the critical time for the SI model is lower bounded by

$$T^c \geq \liminf_{n \to +\infty} \frac{1}{\beta}\sqrt{\frac{\ln n}{2\rho(\mathcal{A}_n)}}(1 - e^{-\sqrt{\frac{\ln n}{2\rho(\mathcal{A}_n)}}}). \tag{23}$$

If $\rho(\mathcal{A}_n) = o(\ln n)$ (e.g. for sparse networks with a maximum degree in $O(1)$), the critical time resumes to $T_c \geq \liminf_{n \to +\infty} \frac{1}{\beta}\sqrt{\frac{\ln n}{2\rho(\mathcal{A}_n)}}$. However, when the graph is denser and $\rho(\mathcal{A}_n)/\ln n \to +\infty$, then $T_c \geq \liminf_{n \to +\infty} \frac{\ln n}{2\beta\rho(\mathcal{A}_n)}$.

## 4.4 Discrete-time Information Cascade

A final example is the discrete-time contagion in which a node infected at time $t$ makes a unique attempt to infect its neighbors at a time $t + T_0$. This defines the *Information Cascade model*, the

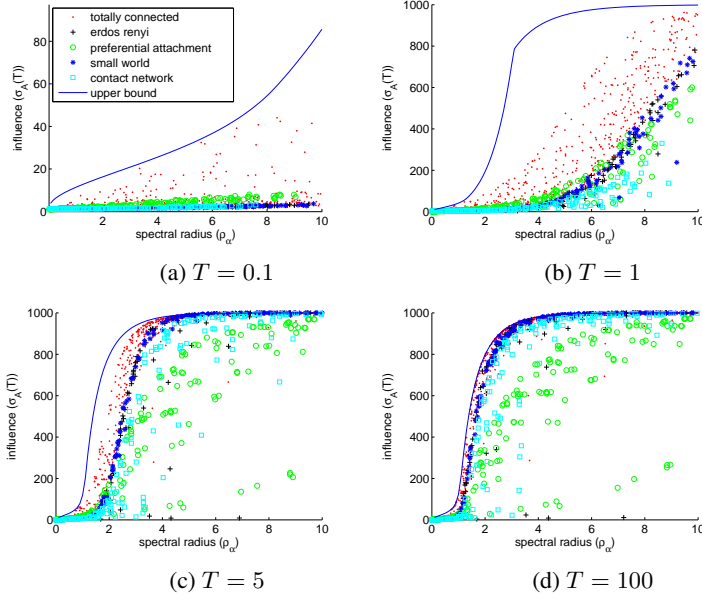

Figure 1: Empirical maximum influence w.r.t. the spectral radius $\rho_\alpha$ defined in Section 4.2 for various network types. Simulation parameters: $n = 1000$, $n_0 = 1$ and $\lambda = 1$.

discrete-time diffusion model studied by the first works on influence maximization [2, 19, 3, 4]. In this setting, $p_{ij}(t) = \alpha_{ij}\delta_{T_0}(t)$ where $\delta_{T_0}$ is the Dirac distribution centered at $T_0$. The spectral radii are given by

$$\rho(s) = \rho_\alpha e^{-sT_0}, \tag{24}$$

and the influence bound of Corollary 1 simplifies to:

**Corollary 5.** *Let $\rho_\alpha \geq 1$, or else $T \leq \frac{T_0}{2(1-\rho_\alpha)}$. If $T < T_0$, then $\sigma_A(T) = n_0$. Otherwise,*

$$\sigma_A(T) \leq n_0 + \sqrt{n_0(n-n_0)}\sqrt{\frac{2eT}{T_0}}\rho_\alpha^{\frac{T}{T_0}}. \tag{25}$$

Moreover, the critical time is lower bounded by

$$T^c \geq \liminf_{n \to +\infty} \frac{\ln n}{2\ln \rho_\alpha}T_0. \tag{26}$$

A notable difference from the exponential transmission probabilities is that $T^c$ is here inversely proportional to $\ln \rho_\alpha$, instead of $\rho_\alpha$ in Eq. 4.2, which implies that, for the same long-term influence, a discrete-time contagion will explode much slower than one with a constant infection rate. This is probably due to the existence of very small infection times for contagions with exponential transmission probabilities.

## 5 Experimental results

This section provides an experimental validation of our bounds, by comparing them to the empirical influence simulated on several network types. In all our experiments, we simulate a contagion with exponential transmission probabilities (see Section 4.2) on networks of size $n = 1000$ and generated random networks of 5 different types (for more information on the respective random generators, see e.g [20]): Erdös-Rényi networks, preferential attachment networks, small-world networks, geometric random networks ([21]) and totally connected networks with fixed weight $b \in [0, 1]$ except for the ingoing and outgoing edges of a single node having, respectively, weight 0 and $a > b$. The reason for simulating on such totally connected networks is that the influence over these networks tend to match our upper bounds more closely, and plays the role of a best case

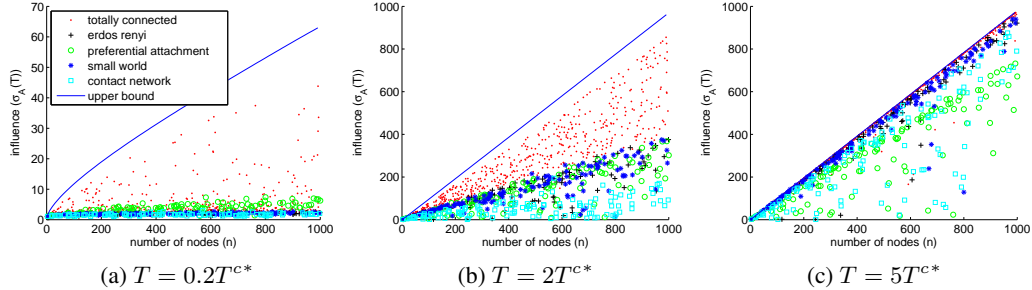

(a) $T = 0.2T^{c*}$        (b) $T = 2T^{c*}$        (c) $T = 5T^{c*}$

Figure 2: Empirical maximum influence w.r.t. the network size for various network types. Simulation parameters: $n_0 = 1$, $\lambda = 1$ and $\rho_\alpha = 4$. In such a setting, $T^{c*} = \frac{\ln n}{2(\rho_\alpha - 1)\lambda} = 1.15$. Note the sub-linear (a) versus linear behavior (b and c).

scenario. More precisely, the transmission probabilities are of the form $p_{ij}(t) = \alpha e^{-t}$ for each edge $(i, j) \in \mathcal{E}$, where $\alpha \in [0, 1[$ (and $\lambda = 1$ in the formulas of Section 4.2).

We first investigate the tightness of the upper bound on the maximum influence given in Proposition 2. Figure 1 presents the empirical influence w.r.t. $\rho_\alpha = -\ln(1 - \alpha)\rho(\mathcal{A})$ (where $\mathcal{A}$ is the adjacency matrix of the network) for a large set of network types, as well as the upper bound in Proposition 2. Each point in the figure corresponds to the maximum influence on one network. The influence was averaged over 100 cascade simulations, and the best influencer (i.e. whose influence was maximal) was found by performing an exhaustive search. Our bounds are tight for all values of $T \in \{0.1, 1, 5, 100\}$ for totally connected networks in the sub-critical regime ($\rho_\alpha < 1$). For the super-critical regime ($\rho_\alpha > 1$), the behavior in $T$ is very instructive. For $T \in \{0.1, 5, 100\}$, we are tight for most network types when $\rho_\alpha$ is high. For $T = 1$ (the average transmission time for the $(\tau_{ij})_{(i,j) \in \mathcal{E}}$), the maximum influence varies a lot across different graphs. This follows the intuition that this is one of the times where, for a given final number of infected node, the local structure of the networks will play the largest role through precise temporal evolution of the infection. Because $\rho_\alpha$ explains quite well the final size of the infection, this discrepancy appears on our graphs at $\rho_\alpha$ fixed. While our bound does not seem tight for this particular time, the order of magnitude of the explosion time is retrieved and our bounds are close to optimal values as soon as $T = 5$.

In order to further validate that our bounds give meaningful insights on the critical time of explosion for super-critical graphs, Figure 2 presents the empirical influence with respect to the size of the network $n$ for different network types and values of $T$, with $\rho_\alpha$ fixed to $\rho_\alpha = 4$. In this setting, the critical time of Corollary 3 is given by $T^{c*} = \frac{\ln n}{2(\rho_\alpha - 1)\lambda} = 1.15$. We see that our bounds are tight for totally connected networks for all values of $T \in \{0.2, 2, 5\}$. Moreover, the accuracy of critical time estimation is proved by the drastic change of behavior around $T = T^{c*}$, with phase transitions having occurred for most network types as soon as $T = 5T^{c*}$.

# 6 Conclusion

In this paper, we characterize the phase transition in continuous-time information cascades between their sub-critical and super-critical behavior. We provide for the first time general influence bounds that apply for any time horizon, graph and set of influencers. We show that the key quantities governing this phenomenon are the spectral radii of given *Laplace Hazard matrices*. We prove the pertinence of our bounds by deriving the first results of this type in several application fields. Finally, we provide experimental evidence that our bounds are tight for a large family of networks.

**Acknowledgments**

This research is part of the SODATECH project funded by the French Government within the program of "*Investments for the Future – Big Data*".

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
