[Supplementary Material · proofs_DynamicInfluenceBounds_NIPS2015.pdf]

## Mathematical Arguments

### Critical time definition: proofs of Theorem 1 and Proposition 1

*Proof of Theorem 1.* Let $S = \{T \in \mathbb{R}_+ | \sigma_n(T) = o(n)\}$. $S$ is an interval containing $0$ since $\sigma_n(0) = 0$ and, if $T \in S$, then $\forall T' \leq T$, $\sigma_n(T') \leq \sigma_n(T)$ and $T' \in S$. Thus $S$ is of the form $[0, T^c[$ or $[0, T^c]$, and let $T^c = \sup S$ (where $T^c \in \mathbb{R} \cup \{+\infty\}$).

For all time sequences $(T_n)_{n \in \mathbb{N}}$ such that $\limsup_{n \to +\infty} T_n < T^c$, $\exists T < T^c$ and $n' \geq 0$ s.t., $\forall n \geq n', T_n \leq T$. Hence, by definition of $T^c$, $\sigma_n(T_n) \leq \sigma_n(T) = o(n)$.

Conversely, if $\sigma_n(T_n) = o(n)$, then $\liminf_{n \to +\infty} T_n \in S$, and $\liminf_{n \to +\infty} T_n \leq T^c$.

Now let $T^{c'}$ verify the two constraints of Theorem 1. The first constraint implies that $\forall T < T^{c'}$, $T \in S$ and $T \leq T^c$, which leads to $T^{c'} \leq T^c$. Moreover, $\forall T < T^c$, $T \in S$ by definition of $T^c$, and $T \leq T^{c'}$ using the second constraint. As a result, $T^{c'} = T^c$ and the critical time is unique. $\square$

*Proof of Proposition 1.* Let $(T_n)_{n \in \mathbb{N}}$ be such that $\liminf_{n \to +\infty} T_n > T^c$. Then $\exists T > T^c$ and $n' \geq 0$ s.t. $\forall n \geq n'$, $T_n \geq T$. However, $T \notin S$ and $\liminf_{n \to +\infty} \sigma_n(T)/n > 0$, which directly implies that $\liminf_{n \to +\infty} \sigma_n(T_n)/n \geq \liminf_{n \to +\infty} \sigma_n(T)/n > 0$.

Conversely, if $(T_n)_{n \in \mathbb{N}}$ is such that $\liminf_{n \to +\infty} \sigma_n(T_n)/n > 0$, then $\limsup_{n \to +\infty} T_n \notin S$ and $\limsup_{n \to +\infty} T_n \geq T^c$. $\square$

### Upper bound on the influence: proofs of Proposition 2 and Corollary 1

Let $\tau_i \in \mathbb{R}_+ \cup \{+\infty\}$ be the infection time of node $i$, and $\tau_{ij} \in \mathbb{R}_+ \cup \{+\infty\}$ the transmission time from node $i$ to node $j$. Let $A \subset \mathcal{V}$ be a set of influencers, i.e. nodes that are infected at time $0$: $\forall i \in A, \tau_i = 0$. Due to the infection dynamic of CTIC, a node $i \notin A$ is infected when at least one of its neighbors is infected, and the respective ingoing edge transmitted the contagion. We thus have the following equation relating infection times $\tau_i$ and $\tau_{ji}$ (see for example [1]): $\forall i \notin A$,

$$\tau_i = \min_{j \in \mathcal{V}} \tau_j + \tau_{ji}. \tag{1}$$

Let $X_i(t) = \mathbb{1}_{\{\tau_i < t\}}$ be the infection state of node $i$ at time $t$. Eq. 1 implies the following equation: $\forall t > 0$ and $i \notin A$,

$$X_i(t) = 1 - \prod_{j \in \mathcal{V}} \left(1 - \mathbb{1}_{\{\tau_j + \tau_{ji} < t\}}\right). \tag{2}$$

We now develop the proofs for Proposition 2 and Corollary 1, which rely on upper bounding the Laplace transform of $\sigma_A(T)$.

**Lemma 1.** *Define $\rho(s) = \rho(\frac{\mathcal{H}(s) + \mathcal{H}(s)^\top}{2})$. Then, for any $A$ such that $|A| = n_0 < n$, denoting by $\hat{\sigma}_A(s) = \int_0^{+\infty} \sigma_A(t) e^{-st} dt$ the Laplace transform of the expected number of nodes reached by the cascade starting from $A$ at time $T$:*

$$s\hat{\sigma}_A(s) \leq n_0 + \gamma(s)(n - n_0), \tag{3}$$

*where $\gamma(s)$ is the smallest solution in $[0, 1]$ of the following equation:*

$$\gamma(s) - 1 + \exp\left(-\rho(s)\gamma(s) - \frac{\rho(s)n_0}{\gamma(s)(n - n_0)}\right) = 0. \tag{4}$$

This result requires two intermediate lemmas: Lemma 2, that proves for $i \in \mathcal{V}$ and $t > 0$ a positive correlation between the events '*node $j$ did not infect node $i$ before time $t$*' and Lemma 4, that bounds the probability that a given node gets infected before $t$.

**Lemma 2.** $\forall i \notin A$ and $t > 0$, $\{1 - \mathbb{1}_{\{\tau_j + \tau_{ji} < t\}}\}_{j \in \mathcal{V}}$ *are positively correlated.*

*Proof.* Denoting by $\mathcal{Q}_i$ the collection of directed paths in $G$ from the influencers $A$ to node $i$, we get the following expression for variables $(\tau_i)_{i \in \mathcal{V}}$ [1]:

$$\tau_i = \min_{q \in \mathcal{Q}_i} \sum_{(j,l) \in q} \tau_{jl} \tag{5}$$

Therefore, for all $i \notin A$ and $t > 0$, the functions $f_{ij}(\tau_{kl})_{(k,l) \in \mathcal{E}} = \{1 - \mathbb{1}_{\{\tau_j + \tau_{ji} < t\}}\}_{j \in \mathcal{V}}$ are increasing with the partial order on $(\tau_{kl})_{(k,l) \in \mathcal{E}}$. We will then make use of the FKG inequality [2] :

**Lemma 3.** *(FKG inequality) Let $L$ be a finite distributive lattice, and $\mu$ a nonnegative function on $L$, such that, for any $(x, y) \in L^2$,*

$$\mu(x \vee y)\mu(x \wedge y) \leq \mu(x)\mu(y) \tag{6}$$

*Then, for any non-decreasing function $f$ and $g$ on $L$*

$$\left( \sum_{x \in L} f(x)g(x) \right) \left( \sum_{x \in L} \mu(x) \right) \geq \left( \sum_{x \in L} f(x)\mu(x) \right) \left( \sum_{x \in L} g(x)\mu(x) \right) \tag{7}$$

Due to the independence of $(\tau_{kl})_{(k,l) \in \mathcal{E}}$, the condition in Lemma 3 is met by their joint distribution, which is a product measure on the product space $\mathbb{R}^{\mathcal{E}}$. Lemma 2 is then obtained by applying Lemma 3 to any couple of functions $(f_{ij}, f_{ik})_{(i,j) \in \mathcal{E}, (i,k) \in \mathcal{E}}$. More specifically, in our problem setting, $L$ is the set of all $(\tau_{kl})_{(k,l) \in \mathcal{E}}$, $\mu(x) = \prod_{(k,l) \in \mathcal{E}} \mathbb{P}(\tau_{kl} = t_{kl})$ is the joint probability distribution of the $\tau_{kl}$ when $x = (t_{kl})_{(k,l) \in \mathcal{E}}$.

$\square$

We then show the following lemma that reveals an implicit inequation satisfied by the $X_i$.

**Lemma 4.** *For all $(i, j) \in \mathcal{V}^2$, let $p_{ij}$ be an integrable function such that $\int_0^{+\infty} p_{ij}(t)dt < 1$. For any $A$ such that $|A| = n_0 < n$ and for any $i \notin A$, the probability $\mathbb{E}[X_i(t)]$ that node $i$ will be reached by the contagion originating from $A$ verifies:*

$$\mathbb{E}[X_i(t)] \leq 1 - \exp\left( -\sum_j (\mathcal{H}_{ji} * \mathbb{E}[X_j])(t) \right), \tag{8}$$

*where $(f * g)(t) = \int f(s)g(t-s)ds$ stands for the convolution of $f$ with $g$ and $\mathcal{H}_{ji}(t) = \frac{\ln(1 - \int_0^{+\infty} p_{ji}(s)ds)}{\int_0^{+\infty} p_{ji}(s)ds} p_{ji}(t)$.*

*Proof.* Eq. 2 and the positive correlation of $\{1 - \mathbb{1}_{\{\tau_j + \tau_{ji} < t\}}\}_{j \in \{1,...,N\}}$ (Lemma 2) imply that

$$\mathbb{E}[X_i(t)] = 1 - \mathbb{E}[\prod_j (1 - \mathbb{1}_{\{\tau_j + \tau_{ji} < t\}})] \leq 1 - \prod_j \mathbb{E}[1 - \mathbb{1}_{\{\tau_j + \tau_{ji} < t\}}] \tag{9}$$

which leads to

$$\begin{aligned} \mathbb{E}[X_i(t)] &\leq 1 - \prod_j \left( 1 - \mathbb{E}[\mathbb{1}_{\{\tau_j + \tau_{ji} < t\}}] \right) \\ &= 1 - \prod_j \left( 1 - \mathbb{E}[\mathbb{E}[X_j(t - \tau_{ji})|\tau_{ji}]] \right), \\ &= 1 - \prod_j \left( 1 - \int_0^{+\infty} \mathbb{E}[X_j(s)]p_{ji}(t-s)ds \right), \end{aligned} \tag{10}$$

since $\forall i, j \in \mathcal{V}$, $\tau_j$ and $\tau_{ji}$ are independent and $p_{ji}$ is the probability density of $\tau_{ji}$. Note that, in our setting, we consider that influencer nodes are infected at time 0, and thus are not infectious before $t = 0$. We then linearize the product in Eq. 10:

$$\begin{aligned} \mathbb{E}[X_i(t)] &\leq 1 - \exp\left( \sum_j \ln(1 - \int_0^{+\infty} \mathbb{E}[X_j(s)]p_{ji}(t-s)ds) \right) \\ &\leq 1 - \exp\left( \sum_j \frac{\ln(1 - \int_0^{+\infty} p_{ji}(s)ds)}{\int_0^{+\infty} p_{ji}(s)ds} \int_0^{+\infty} \mathbb{E}[X_j(s)]p_{ji}(t-s)ds \right) \\ &= 1 - \exp\left( -\sum_j (\mathcal{H}_{ji} * \mathbb{E}[X_j])(t) \right), \end{aligned} \tag{11}$$

since we have on the one hand, for any $x \in [0,1]$ and $a < 1$, $\ln(1 - ax) \geq \ln(1 - a)x$ (in Eq. 11, we chose $a = \int_0^{+\infty} p_{ji}(s)ds$ and $x = \frac{\int_0^{+\infty} \mathbb{E}[X_j(s)]p_{ji}(t-s)ds}{\int_0^{+\infty} p_{ji}(s)ds}$), and on the other hand $\mathcal{H}_{ji}(t) = \frac{\ln(1 - \int_0^{+\infty} p_{ji}(s)ds)}{\int_0^{+\infty} p_{ji}(s)ds} p_{ji}(t)$ by definition of $\mathcal{H}$. Note that $\frac{\ln(1 - \int_0^{+\infty} p_{ji}(s)ds)}{\int_0^{+\infty} p_{ji}(s)ds}$ is approximately 1 when $\int_0^{+\infty} p_{ji}(s)ds$ is close to 0. $\qquad\square$

*Proof of Lemma 1.* From here, Proposition 1 follow from Lemma 4 in the exact same way than, in [3], the proof of Proposition 1 is deduced from Lemma 8. However, we give here the fully detailed proof for sake of completeness.

Let $f_i(s) = \int_0^{+\infty} \mathbb{E}[X_i(t)]se^{-st}dt$, then, using Jensen's inequality, $\forall i \notin A$ and $s \geq 0$,

$$f_i(s) \leq 1 - \exp\left( - \sum_j \mathcal{L}\mathcal{H}_{ji}(s)f_j(s) \right), \tag{12}$$

where $\mathcal{L}\mathcal{H}_{ji}(s) = \int_0^{+\infty} \mathcal{H}_{ji}(t)e^{-st}dt$ is the Laplace transform of $\mathcal{H}_{ji}$. Note also that $\forall i \in A, f_i(s) = 1$.

For every $i \in [1...n]$, we define $Z_i = \big(f_i(s)\big)_i$ and the vector $Z = (Z_i)_{i \in [1...n]}$. Using lemma 4 and convexity of exponential function, we have for any $u \in R^n$ such that $\forall i \in A, u_i = 0$ and $\forall i \notin A, u_i \geq 0$,

$$u^\top Z \leq |u|_1 \left( 1 - \sum_{i=1}^{n-1} \frac{u_i}{|u|_1} \exp(-(\mathcal{L}\mathcal{H}^\top Z)_i) \right) \leq |u|_1 \left( 1 - \exp\left( - \frac{Z^\top \mathcal{L}\mathcal{H}u}{|u|_1} \right) \right) \tag{13}$$

where $|u|_1 = \sum_i |u_i|$ is the $L_1$-norm of $u$.

Now taking $u = (1_{i \notin A}Z_i)_i$ and noting that $\forall i, u_i \leq Z_i$, we have

$$\frac{Z^\top Z - n_0}{|Z|_1 - n_0} \leq 1 - \exp\left( - \frac{Z^\top \mathcal{L}\mathcal{H}Z}{|Z|_1 - n_0} \right) \leq 1 - \exp\left( - \frac{\rho(s)(Z^\top Z - n_0)}{|Z|_1 - n_0} - \frac{\rho(s)n_0}{|Z|_1 - n_0} \right) \tag{14}$$

where $\rho(s) = \rho(\frac{\mathcal{L}\mathcal{H} + \mathcal{L}\mathcal{H}^\top}{2})$. Defining $y = \frac{Z^\top Z - n_0}{|Z|_1 - n_0}$ and $z = |Z|_1 - n_0 = s\hat{\sigma}_A(s) - n_0$, the inequation above rewrites

$$y \leq 1 - \exp\left( - \rho(s)y - \frac{\rho(s)n_0}{z} \right) \tag{15}$$

But by Cauchy-Schwarz inequality applied to $u$, $(n - n_0)(Z^\top Z - n_0) \geq (|Z|_1 - n_0)^2$, which means that $z \leq y(n - n_0)$. We now consider the equation

$$x - 1 + \exp\left( - \rho(s)x - \frac{\rho(s)n_0}{x(n - n_0)} \right) = 0 \tag{16}$$

Because the function $f : x \to x - 1 + \exp\left( - \rho(s)x + \frac{\rho(s)n_0}{x(n-n_0)} \right)$ is continuous, verifies $f(1) > 0$ and $\lim_{x \to 0^+} f(x) = -1$, equation 16 admits a solution $\gamma(s)$ in $]0, 1[$.

We then prove by contradiction that $z \leq \gamma(s)(n - n_0)$. Let us assume $z > \gamma(s)(n - n_0)$. Then $y \leq 1 - \exp\left( -\rho(s)y - \frac{\rho(s)n_0}{\gamma(s)(n-n_0)} \right)$. But the function $h : x \to x - 1 + \exp\left( -\rho(s)x + \frac{\rho(s)n_0}{\gamma(s)(n-n_0)} \right)$ is convex and verifies $h(0) < 0$ and $h(\gamma(s)) = 0$. Therefore, for any $y > \gamma_1$, $0 = f(\gamma_1) \leq \frac{\gamma(s)}{y}f(y) + (1 - \frac{\gamma(s)}{y})f(0)$, and therefore $f(y) > 0$. Thus, $y \leq \gamma(s)$. But $z \leq y(n - n_0) \leq \gamma(s)(n - n_0)$ which yields the contradiction. $\qquad\square$

Using Lemma 1, we may now prove Proposition 2:

*Proof of Proposition 2.* $\forall s \geq 0$, $T \geq 0$ and $t \geq 0$, $e^{-st} \geq e^{-sT}1_{\{t < T\}}$, hence, using Lemma 1, $s\hat{\sigma}_A(s) = \sum_i \mathbb{E}[e^{-s\tau_i}] \geq n_0 + (\sigma_A(T) - n_0)e^{-sT}$ which leads to the desired inequality. $\qquad\square$

*Proof of Corollary 1.* Using Eq. 16 and the fact that $1 - e^{-x} \leq x$, we get $\gamma(s) \leq \rho(s)\gamma(s) + \frac{\rho(s)n_0}{\gamma(s)(n-n_0)}$ which rewrites $\gamma(s) \leq \sqrt{\frac{\rho(s)n_0}{(1-\rho(s))(n-n_0)}}$ in the case $\rho(s) < 1$. Therefore,

$$\sigma_A(T) \leq n_0 + \sqrt{n_0(n-n_0)} \min_{\{s \geq 0 | \rho(s) < 1\}} \left( \sqrt{\frac{\rho(s)}{1-\rho(s)}} e^{sT} \right). \tag{17}$$

$\square$

**Upper bounds on the critical time: proofs of Corollary 2 and Corollary 3**

*Proof of Corollary 2.* Since $e^{-st}$ is decreasing w.r.t. $s$, $\mathcal{LH}_{ij}(s)$ is decreasing. Thus, the Perron-Frobenius theorem implies that $\rho(s)$ is decreasing. When $\rho(0) \geq 1$, $\rho^{-1}(1 - \epsilon)$ exists and is uniquely defined, and using Corollary 1 and 2, $\sigma_A(T) \leq n_0 + (n-n_0)\gamma(\rho^{-1}(1-\epsilon))e^{\rho^{-1}(1-\epsilon)T} \leq n_0 + \sqrt{\frac{n_0(n-n_0)}{\epsilon}}e^{\rho^{-1}(1-\epsilon)T}$. $\square$

*Proof of Corollary 3.* If $\limsup_{n \to +\infty} \frac{2\rho^{-1}(1)T_n}{\ln n} < 1$, then $\exists \alpha > 0$ and $n' \geq 0$ s.t. $\forall n \geq n'$, $\rho^{-1}(1)T_n \leq \frac{(1-\alpha)\ln n}{2}$. Furthermore, $\lim_{n \to +\infty} \frac{\rho^{-1}(1-\frac{1}{\ln n})}{\rho^{-1}(1)} = 1$, thus $\exists n'' \geq n'$ s.t. $\forall n \geq n''$, $\rho^{-1}(1 - \frac{1}{\ln n}) \leq \frac{1-\alpha/2}{1-\alpha}\rho^{-1}(1)$. Using Corollary 2 with $\epsilon = \frac{1}{\ln n}$, $\sigma_A(T) \leq 1 + \sqrt{\ln n}(n-1)e^{\rho^{-1}(1-\frac{1}{\ln n})T} \leq 1 + \sqrt{\ln n}n^{1-\alpha/4} = o(n)$. $\square$

**Application to particular contagion model: proofs of Corollary 4, Proposition 3, Proposition 5 and Corollary 5**

*Proof of Corollary 4.* Taking $\rho(s) = \frac{\lambda}{\lambda+s}\rho_\alpha$, Corollary 1 rewrites

$$\sigma_A(T) \leq n_0 + \sqrt{n_0(n-n_0)} \min_{s \geq 0} \left( \sqrt{\frac{\lambda}{s + \lambda(1-\rho_\alpha)}} e^{sT} \right). \tag{18}$$

The function $f(s) = \sqrt{\frac{\lambda}{s+\lambda(1-\rho_\alpha)}}e^{sT}$ admits a unique minimum in $s_{min} = \frac{1}{2T} + \lambda(\rho_\alpha - 1)$. The minimum for $s \geq 0$ is therefore met for $s = s_{min}$ if $\lambda T(1 - \rho_\alpha) < \frac{1}{2}$ and $s = 0$ otherwise. The results follow immediately. $\square$

*Proof of Proposition 3.* In order to prove Proposition 3, it is sufficient to show that Lemma 4 still holds for the SIR model, with $p_{ij}(t) = \beta \exp(-(\delta + \beta)t)$ for $(i,j) \in \mathcal{E}$. For $i \in \mathcal{V}$, let $\theta_i$ be the random removal time of node $i$. Infection times $\tau_i$ are then given by the following expression, where $\mathcal{Q}_i$ is the collection of directed paths in $G$ from the influencers $A$ to node $i$:

$$\tau_i = \min_{q \in \mathcal{Q}_i} \sum_{(j,l) \in q} \tau_{jl} \mathbb{1}_{\{\tau_{jl} < \theta_j\}} \tag{19}$$

Therefore $\forall i \notin A$ and $t > 0$, the functions $f_{ij}(\tau, \theta) = \{1 - \mathbb{1}_{\{\tau_j + \tau_{ji} < t\}}\mathbb{1}_{\{\tau_{ji} < \theta_j\}}\}_{j \in \mathcal{V}}$ are increasing with respect to the partial order on $\mathbb{R}^{\mathcal{E}} \times \mathbb{R}^{\mathcal{V}}$ defined for any $X^1 = (\tau_1^1, ... \tau_m^1, \theta_1^1 ... \theta_n^1) \in \mathbb{R}^{\mathcal{E}} \times \mathbb{R}^{\mathcal{V}}$ and $X^2 = (\tau_1^2, ... \tau_m^2, \theta_1^2 ... \theta_n^2) \in \mathbb{R}^{\mathcal{E}} \times \mathbb{R}^{\mathcal{V}}$ by:

$$X^1 \geq X^2 \iff \begin{cases} \tau_{ij}^1 \geq \tau_{ij}^2 & \text{for any } (i,j) \in \mathcal{E} \\ \theta_i^1 \leq \theta_j^2 & \text{for any } i \in \mathcal{V} \end{cases}. \tag{20}$$

Variables $(\tau_{ij})_{(i,j) \in \mathcal{E}}$ and $(\theta_i)_{i \in \mathcal{V}}$ being independent, we can still apply FKG inequality (Lemma 3) and deduce the positive correlation, for any $i \notin A$ and $t > 0$, of the random variables $\{1 - \mathbb{1}_{\{\tau_j + \tau_{ji} < t\}}\mathbb{1}_{\{\tau_{ji} < \theta_j\}}\}_{j \in \mathcal{V}}$. We then introduce, for any $(i,j) \in \mathcal{E}$:

$$\overline{\tau_{ji}} = \begin{cases} \tau_{ji} & \text{if } \tau_{ji} < \theta_j \\ +\infty & \text{if } \tau_{ji} \geq \theta_j \end{cases}. \tag{21}$$

It is straightforward that each $\overline{\tau_{ji}}$ is a random variable over $\mathbb{R}_+ \cup \{+\infty\}$ with probability distribution $p_{ij}$, and that $\overline{\tau_{ji}}$ is independent of $\tau_j$. We also have, for any $i \notin A$, $t > 0$ and $(i,j) \in \mathcal{E}$:

$$\{1 - \mathbb{1}_{\{\tau_j + \tau_{ji} < t\}} \mathbb{1}_{\{\tau_{ji} < \theta_j\}}\} = \{1 - \mathbb{1}_{\{\tau_j + \overline{\tau_{ji}} < t\}}\} \tag{22}$$

Lemma 4 for the SIR case (and therefore Proposition 3 and its subsequent corollaries) are then proved from following the same steps than in the independent transmission events case, except replacing $(\tau_{ji})_{(i,j)\in\mathcal{E}}$ by $(\overline{\tau_{ji}})_{(i,j)\in\mathcal{E}}$ □

*Proof of Proposition 5.* $\rho(s) = \frac{\beta T_n}{1 - e^{-\beta T_n}} \frac{\beta}{\beta + s}(1 - e^{-(\beta + s)T_n})\rho(\mathcal{A}) \leq \frac{\beta^2 T_n \rho(\mathcal{A})}{(1 - e^{-\beta T_n})s}$, which implies $\rho^{-1}(1)T_n \leq \frac{(\beta T_n)^2 \rho(\mathcal{A})}{1 - e^{-\beta T_n}}$. Let $f(x) = \frac{x^2}{1 - e^{-x}}$, $f$ is increasing and $\forall a \geq 0$, $f(x) = a \implies x \geq \sqrt{a}(1 - e^{-\sqrt{a}})$. Hence, if $\limsup_{n \to +\infty} \frac{\beta T_n}{\sqrt{\frac{\ln n}{2\rho(\mathcal{A}_n)}}(1 - e^{-\sqrt{\frac{\ln n}{2\rho(\mathcal{A}_n)}}})} < 1$, then $\exists \alpha > 0$ s.t. $\beta T_n \leq (1-\alpha)\sqrt{\frac{\ln n}{2\rho(\mathcal{A}_n)}}(1 - e^{-\sqrt{\frac{\ln n}{2\rho(\mathcal{A}_n)}}})$, and the concavity of $1 - e^{-x}$ implies that $\beta T_n \leq \sqrt{\frac{(1-\alpha)\ln n}{2\rho(\mathcal{A}_n)}}(1 - e^{-\sqrt{\frac{(1-\alpha)\ln n}{2\rho(\mathcal{A}_n)}}})$. Finally, $f(\beta T_n) \leq \frac{(1-\alpha)\ln n}{2\rho(\mathcal{A}_n)}$ and $\frac{2\rho^{-1}(1)T_n}{\ln n} \leq 1 - \alpha$. Applying Corollary 3 proves the desired result. □

*Proof of Corollary 5.* Taking $\rho(s) = \rho_\alpha e^{-sT_0}$, Corollary 1 rewrites

$$\sigma_A(T) \leq n_0 + \sqrt{n_0(n - n_0)} \min_{s \geq 0} \left( \sqrt{\frac{\rho_\alpha e^{-sT_0}}{1 - \rho_\alpha e^{-sT_0}}} e^{sT} \right). \tag{23}$$

and $s = \frac{1}{T_0}\left(\ln \rho_\alpha - \ln(1 - \frac{T_0}{2T})\right)$ gives

$$\sigma_A(T) \leq n_0 + \sqrt{n_0(n - n_0)} \sqrt{\frac{2T}{T_0} - 1} \left( \frac{\rho_\alpha}{1 - \frac{T_0}{2T}} \right)^{\frac{T}{T_0}}. \tag{24}$$

The final result follows by upper bounding $\left(1 - \frac{T_0}{2T}\right)^{\frac{1}{2} - \frac{T}{T_0}}$ by $\sqrt{e}$ due to the monotonic increase of $x \to (x-1)\ln(1 - \frac{1}{x})$ on $[1, +\infty[$ and its limit when $x \to +\infty$. □

## Additional references

[1] Nan Du, Le Song, Manuel Gomez-Rodriguez, and Hongyuan Zha. Scalable influence estimation in continuous-time diffusion networks. In *Advances in Neural Information Processing Systems*, pages 3147–3155, 2013.

[2] Cees M Fortuin, Pieter W Kasteleyn, and Jean Ginibre. Correlation inequalities on some partially ordered sets. *Communications in Mathematical Physics*, 22(2):89–103, 1971.

[3] Remi Lemonnier, Kevin Scaman, and Nicolas Vayatis. Tight bounds for influence in diffusion networks and application to bond percolation and epidemiology. In *Advances in Neural Information Processing Systems*, pages 846–854, 2014.