[Reviews · NeurIPS 2015]

Submitted by Assigned_Reviewer_1

This paper extends NIPS'14 paper [17] to upper bound the maximum influence and lower bound the time on the critical time of diffusion over networks. The particular insight is to use the Laplacian Hazard matrix (containing the laplace transform of the transmission rate function) as an extension of the harzard matrix in [17], and relate the size of influence (# of infected nodes) to the spectral radius of this matrix.

The authors have also shown a few specific cases of particular transmission probabilities over time, e.g. contant, exponential and SI/SIR.

The insight and the derived bounds are novel. The paper is clearly laid out.

I have not checked the accompanying proofs, they seem reasonable. It is also likely that I did not understood parts of the argument, hence the low confidence rating.
Summary: This paper extends NIPS'14 paper [17] to upper bound the maximum influence and lower bound the time on the critical time of diffusion over networks. This is a theoretical contribution that is novel, the influence bounds are shown to be tight visa simulation, and potentially applicable to algorithm and experimentation (not shown in this paper).

Submitted by Assigned_Reviewer_2

- It would to point out that p_ij(t) are actually subprobabilities, since \int_{0}^{\infty} p_{ij}(t) < 1.

- Daneshmand et al. [ICML '14] also use the naming "hazard matrix" to denote a different quantity on a set of observed cascades. It would be useful to point out this, highlighting the differences.

- In section 4.1., the authors do not explicitly state that \alpha_{ij} in [0, 1). I think this is necessary for p_{ij}(t) to be a (sub)probability.

- The writing of the first paragraph of the introduction could be heavily improved.

- The authors refer to [3] as early work on influence maximization. [3] does not tackle the influence maximization problem, but a different one.

- The proof of Lemma 3 could be described more explicitly. In particular, explicitly relate f(x), g(x) and \mu(x) to the actual functions in the problem setting. This would help a reader to understand the definition of "right positive correlation", which is never formally written.

- In Eq. (11) in the proof of Lemma 4, the authors should explicitly write what is a and x in ln(1 - ax) >= ln (1-a)x to help the reader.

- In Eq. 12, should f_i(s) on the right side not be f_j(s)?

- There is some very recent work that aims to establish the theoretical foundations of influence models. In particular, there are sample complexity results on the recovery of influence models:

Daneshmand et al., ICML 2014 Pouget-Abadie et al., ICML 2015

Although different to this work, it would help the reader understanding that there is already an on-going interest on the theoretical analysis of such models.
Summary: The authors perform a theoretical analysis of diffusion of contagions in networks. In particular, they develop a novel non-asymptotic upper bound on the influence (average number of infected nodes) of any source set in a network and a lower bound on the critical time after which a contagion becomes super critical. The analysis uses the concept of Laplace Hazard matrix, an intuitive quantity, which depends on the network structure and edge probability distributions and is based on previous work. Their bounds are fairly general and the authors instantiate them on several well-known diffusion models. The authors also validate their theoretical results with a fairly brief (and perhaps too shallow) experimental evaluation on synthetic data.

The problem is scientifically interesting and is timely. To the best of my knowledge, there is no previous work on influence bounds for continuous time diffusion networks. The paper is mathematically sound. The experimental results could be a bit more insightful/illustrative.

Submitted by Assigned_Reviewer_3

The authors studied the transition phenomena in information cascades by showing that the spectral radius of Laplace Hazard matrix can characterize the dynamics of the contagion. They focused on the non-asymptotic influence bounds, the super-critical time, and the tight lower bounds of critical explosion time.

Strong points: 1. This work has in-deep theoretical bounds. 2. It focuses on network information cascades, which is a question of great concern to research community. 3. This paper is well organized and easy to read.

Suggestions: Experiments can be improved to verify their theory. The authors evaluated the case of 'exponential transition probabilities'. They can also verify the cases of 'fixed transition pattern' and 'SI and SIR epidemic models'. The tightness in Fig.1 is only for fully connected networks?

Summary: This paper extends the approach published by Lemonnier et al.[17] to deal with anytime influence bounds. The approach has strict theoretical bounds.

Submitted by Assigned_Reviewer_4

Understanding the process influence maximization is critical in several domains like epidemiology,

computer networks, marketing, etc and the most widely used model to capture the temporal behavior in the cascade process is the CTIC model. Assessing the result quality is a challenge and prior work shows that the spectral radius of the Hazard matrix plays

a key role and derives a tight bound for the influence in a general graph. The authors extend the prior work to handle anytime influence bounds. They define when a contagion is sub-critical and super-critical and provide a lower bound when the contagion becomes super-critical and show the application to several contagion models. The proofs look comprehensive. The experimental results show the goodness of the model on the synthetic dataset but it would be useful to also see how the model scales up for large real graphs.

Summary: The paper looks like a good paper and is well written. The weak points may be the novelty and lack of evaluation on real datasets.

Author Feedback
Author rebuttal: We thank the reviewers for their positive evaluations and their useful comments. Two main issues were raised:

1. From asymptotics (Lemonnier et al. [17]) to non-asymptotics (this work)
Some reviewers suggest that this is a simple generalization of previous work. Although the framework is similar to the reference [17], the nature of our contributions is radically different and we want to highlight two significant implications of the non-asymptotic results presented in our paper:

A. Application to continuous-time diffusion processes.
Over the past few years, continuous-time information cascades have become the gold standard when dealing with influence maximization problems, acknowledging the key role played by the precise understanding of time-varying transmission functions. Our work provides the first theoretical framework allowing the evaluation of these algorithms with respect to our closed-form bound. We therefore believe this generalization brings the missing brick between the fine theoretical work of Lemonnier et al. and relevance for evaluating state-of-the-art influence maximization algorithms very popular for the NIPS audience.

B. Introduction of a new concept of "critical time".
The study of diffusion processes has always distinguished between supercritical (or explosive) diffusion processes and subcritical processes. Up to our knowledge, this is the first time that a non-trivial estimate on how much time is available for decisioners to react when faced to a supercritical diffusion process is precisely quantified. We believe that this notion is of interest per se and should be further studied in future work.

2. Arguments for our choices in the experimental section
First, we would like to recall that the main goal of this paper was to provide the first anytime influence bounds for the problem of influence maximization. Therefore, the aim of our experimental section was more to show that these bounds were not "unreasonably far" from what can be observed on real graphs than to prove actual tightness for all graphs and excitation functions.
Also note that because the influence estimation problem is NP-hard and our results apply to all graphs and excitation functions, we cannot expect more than to show that we match the bound for some graphs and some excitation functions. This is why we chose to focus on exponential transmission functions and still speak of "tightness" in Fig. 1.
We agree that the precise comparison of our bounds with different influence maximization algorithms for multiple graphs would bring interesting insights on our work and the continuous-time influence maximization problem in general. A natural goal would be to prove that state-of-the-art influence maximization algorithms are close to tightness on graphs were the actual maximum of influence is not known.
However, we believe that these experiments, involving large-scale computations of different influence maximization algorithms, are far beyond the scope of this theoretical paper, but they will definitely be developed in future research.

3. Comments on additional remarks
We will add the mentioned references, and remove [3] from the list of early works on influence maximization. We would also like to thank Reviewer 1 for all the minor corrections provided (\alpha_{ij} in [0,1[, adding insights when using ln(1-ax) >= ln(1-a)x, f_i(s) vs. f_j(s), difference with the notion of Hazard matrix defined in Daneshmand et al. [ICML '14]) and will add them to the final version of the article.

Concerning Lemma 3, we will make clearer that, in our problem setting, L is the set of all (tau_{kl})_{k,l \in {1,...,n}}, mu(x) = prod_{kl} P(tau_{kl} = t_{kl}) is the joint probability distribution of the tau_{kl} when x = (t_{kl})_{k,l \in {1,...,n}}, and f(x) and g(x) are the functions {1 - 1_{tau_j + tau_{ji} < t}} for j in {1,...,n} (and i fixed).

In the formal definition (section 2.1), we state that p_ij(t) is a probability distribution on R\cup{+\infty}. Although we agree that, for particular examples, we define p_{ij}(t) for t\in R and leave to the reader the calculation of p_{ij}(+\infty), we are concerned that mentioning sub-probabilities for p_{ij}(t), although its definition should be on R\cup{+\infty}, may lead to confusion for some readers.

In our experimental section, we mean by "tightness" that, for a fixed value of the spectral radius of the Hazard matrix, there exists a network such that the bound is very close to the true value.